# Nutritional Content According to the Presence of Front of Package Marketing Strategies: The Case of Ultra-Processed Snack Food Products Purchased in Costa Rica

**DOI:** 10.3390/nu11112738

**Published:** 2019-11-12

**Authors:** Tatiana Gamboa-Gamboa, Adriana Blanco-Metzler, Stefanie Vandevijvere, Manuel Ramirez-Zea, Maria F. Kroker-Lobos

**Affiliations:** 1School of Public Health, Universidad de Costa Rica, Montes de Oca, San Jose 11501, Costa Rica; 2Costa Rican Institute of Research and Teaching in Nutrition and Health (INCIENSA), Curridabat, Cartago 4-2250, Costa Rica; ablanco@inciensa.sa.cr; 3Department of Epidemiology and Biostatistics, School of Population Health, University of Auckland, Auckland 1010, New Zealand; s.vandevijvere@auckland.ac.nz; 4INCAP Research Center for the Prevention of Chronic Diseases, Institute of Nutrition of Central America and Panama (INCAP), Guatemala City 1188, Guatemala; mramirez@incap.int (M.R.-Z.); fkroker@incap.int (M.F.K.-L.)

**Keywords:** food labeling, health claims, ultra-processed foods, obesity, children, food environment, Costa Rica

## Abstract

The industry uses nutrition and health claims, premium offers, and promotional characters as marketing strategies (MS). The inclusion of these MS on ultra-processed products may influence child and adolescent purchase behavior. This study determined the proportion of foods carrying claims and marketing strategies, also the proportion of products with critical nutrients declaration, and nutritional profile differences between products that carry or not claims and MS on the front-of-package (FoP) of ultra-processed food products sold in Costa Rica. Data were obtained from 2423 photographs of seven food groups consumed as snacks that were sold in one of the most widespread and popular hypermarket chains in Costa Rica in 2015. Ten percent of products lacked a nutrition facts panel. Sodium was the least reported critical nutrient. Energy and critical nutrients were significantly highest in products that did not include any nutrition or health claim and in products that included at least one MS. Forty-four percent and 10% of all products displayed at least one nutrition or at least one health claim, respectively, and 23% displayed at least one MS. In conclusion, regulations are needed to restrict claims and marketing on ultra-processed food packages to generate healthier food environments and contribute to the prevention of childhood and adolescent obesity in Costa Rica.

## 1. Introduction

North, Central, and South America have the highest prevalence of childhood obesity in the world [1,2]. Specifically, the proportion of Latin American children with excess weight is more than 20% in more than one third of the countries [2]. Costa Rica is no exception, where 34% of children between 6 to 12 years old are overweight or obese [3]. An obesogenic food environment, characterized by ultra-processed food products highly available and advertised, is one of the significant drivers of obesity that could be preventable [4,5,6]. 

Evidence suggests a strong association between the rise of childhood and adolescent obesity and an increase in daily snacking [7]. Snacking is associated with a greater intake of high energy dense ultra-processed food products such as sweetened caloric beverages, salty snacks, breakfast cereals, muffins, cookies, cereal bars, and confectionery [8,9,10,11,12,13]. Snacks are often ultra-processed products containing high calories content, added sugars, sodium, and saturated fat, and a small quantity of fiber and micronutrients [14,15,16].

Food and beverage packages are used as valuable marketing tools to motivate people to buy and consume them [17]. Nutrition and health claims, premium offers, and promotional characters are common marketing strategies (MS) used by the food industry. The front-of-package (FoP) labeling is the most visible part of the package. It displays pictorial and, symbolic elements, and short text claims in a place where they are more likely to receive consumers’ attention [18,19]. The inclusion of these MS on ultra-processed products may influence child and adolescent purchasing requests and food preferences [20,21]. A meta-analysis concluded that health and nutrition claims have substantial effects on dietary choices, but studies agree that it depends on the type of product [22]. 

Additionally, premium offers and promotional characters encourage and motivate children to think that products are healthier and funny than others [23]. Many studies confirm the vulnerability of children because of their inability to understand the persuasive intent or such marketing strategies [24,25,26,27]. These strategies may lead to excessive consumption of ultra-processed foods with the corresponding high intake of added sugars, saturated fats, trans fat, sodium, and calories [20,24,28].

Recommended actions for overweight and obesity prevention in children and adolescents include the improvement of policies and regulations to decrease the consumption of unhealthy foods. FoP labeling and healthy school food environments (food provision in school cafeterias) are among the recommended actions [13]. The Pan American Health Organization (PAHO) has developed a Nutrient Profile to determine the criteria for acceptable amounts of critical nutrients such as salt, added sugars, saturated fats, and trans fats, which can be helpful when designing such policies [14].

Policies that regulate food advertising and FoP labeling have being implemented in different countries [21]. At the regional level, Chile is a pioneer country where all forms of promotion aimed at children under 14 years of age, are prohibited. Moreover, a FoP warning label system has been established to inform consumers when a product exceeds thresholds for sodium, saturated fat, added sugars, and energy [21,28].

In Costa Rica, the food industry follows the Central American Technical Regulation for General Labelling of Food Products and the Central American Technical Regulation of Nutrition Labeling, which is based on the CODEX Alimentarius. However, this last regulation establishes voluntary declaration of critical nutrients such as added sugars, sodium, or fat, except when a health or nutrition claim is displayed, and no effort has been made to implement a base-evidence FoP labeling systems or marketing regulation targeted to children [29,30,31].

A previous study in Costa Rica found that 79% of breakfast cereals and 79% of dairy products displayed a nutrition claim, and 22.3% of products displayed at least one promotional character [21].

Food packages can serve as a tool for consumers to make informed decisions about their food choices. Little is known about the information on ultra-processed food products commonly consumed among children and adolescents in Costa Rica. That is why the results of this research could be valuable evidence for future improvements in the regulation of food labeling practices.

Therefore, this study aims to determine: (1) the proportion of foods carrying claims and marketing strategies, (2) proportion of products with critical nutrients declaration and (3) nutritional profile differences between products that carry or not claims and MS on the FoP of ultra-processed food products sold in Costa Rica.

## 2. Materials and Methods 

### 2.1. Design and Procedure 

This study was a cross-sectional analysis of food label information from a database of photographs of all the food and beverage (*n* = 7953) sold in one of the most widespread and popular supermarket chains in Costa Rica. Since 88% of Costa Rican consumers purchase their foods in supermarkets [32], it is likely that a large proportion of products commonly purchased was captured. The database was done in 2015 as part of the National Salt Reduction Program [33,34]. Trained research assistants visited the supermarket (researchers selected the location by convenience) and photographed all packaged products with a smartphone application designed for this purpose. They included the information in the database system administered by the George Institute for Global Health in Australia [35]. Photographs included FoP, nutrient content, and ingredient information. 

### 2.2. Selection of Food Categories to Be Included in the Analysis

We selected ultra-processed food and beverage products commonly consumed as snacks by children and adolescents, including: non-alcoholic beverages with and without added sugars, savory snacks, sweetened milk beverages, breakfast cereals, muffins, cookies, cakes, pastries, cereal bars and confectionery (Figure 1). These categories were selected because they are energy-dense, low-nutrient foods commonly consumed as snacks by children and adolescents and also due to their association with overweight, obesity [9,10,11,13,14,15,16,36,37]. 

We excluded fresh, culinary ingredients and, minimally processed and processed food products. Food groups excluded were: convenience foods, bread and bakery, edible oils and oil emulsions products, eggs, fruit and vegetables, sauces and spreads, fish and seafood products, processed fish, meat and meat products, sugar, honey, and related products. From 7953 products included in the database, 5496 were excluded because they were not from the selected food groups. Fifty-five additional products were excluded due to unreadable picture information, duplicate products, or missing packages, including a total of 2402 products for the present analysis.

### 2.3. Data and Measures 

Nutritional composition information (energy, total fat, trans fat, saturated fat, added sugars, and sodium) was obtained from the nutritional facts panel (NFP) captured in the pictures of the products. The nutrition and health claims and the presence of marketing strategies such as promotional characters and premium offers were obtained from the FoP label. All variables were assessed and classified according to the standardized international methodology of the International Network for Food and Obesity/NCDs Research, Monitoring and Action Support (INFORMAS) [38].

For claim and MS identification and classification, two research assistants were trained by the first author using the INFORMAS protocol (Appendix A shows examples of the classification of claims and MS). After the training, double data entry was performed in forty randomly selected products. In case of discrepancies, the first author discussed the results with the research assistants to reach a consensus. Cohen’s kappa statistic was used to test interrater reliability. A k value higher than 0.8 was achieved. 

#### 2.3.1. Nutrition and Health Claims

Nutrition claims are any representation, which states, suggests or implies that a food has particular nutritional property including but not limited to the energy value and the content of protein, fat, and carbohydrates, as well as the content of vitamins and minerals [31,38]. Nutrition claims include health-related ingredient claims, nutrient content claims, and nutrient comparison claims. 

Health claims are any representation that states, suggests, or implies that a relationship exists between a food or a constituent of that food and health [31,38]. Health claims include general health claims, nutrient and other function claims and reduction of disease risk claims. 

#### 2.3.2. Marketing Strategies

Marketing strategies were classified as promotional strategies and premium offers. Promotional strategies included: cartoon or company owned character, licensed character, amateur sports person, celebrity, movie tie-in, famous sports person or team, on-sports, historical events or festivals, and characters for kids. Premium offers included: game and app downloads, contests, buy two get one free, 20% extra or similar, limited edition, social charity, gift or collectable.

### 2.4. Declaration of Critical Nutrients and Nutritional Content

The percentage of products that reported critical nutrients for each food category was calculated, estimating the number of products that reported critical nutrients divided into the total number of products per food category. Additionally, we extracted the nutritional content of energy, total fat, saturated fat, trans fat, sugar, and sodium from the NFP.

### 2.5. Statistical Analysis

We estimated the proportion of products reporting critical nutrients in each food category, and compared energy and nutrient density (per 100 g or 100 mL) for products with and without claims and marketing strategies on the FoP. Variables with a non-parametric distribution, such as energy and nutrient composition of food products were expressed in medians and interquartile ranges (25th–75th percentile). Additionally, the total number of claims, claim type and the number of products carrying claims for each claim type was determined. 

The Mann Whitney test (95%CI) was used to determine differences of nutritional content by the presence or not of claims and marketing strategies. The Marascuillo procedure was used to test the differences between the proportion of products that included claims and marketing strategies on the FoP of the seven different food categories. A Chi square test (95%CI) was performed to compare the content of energy and critical nutrients, of products with or without at least one claim or marketing strategy in the FoP. STATA 14.0 (StataCorp, Texas, USA) and Excel 15.13.3 (Microsoft Corporation, Washington, DC, USA) were used for all statistical analyses. We hypothesized that products that displayed claims and marketing strategies have higher critical nutrient content than the rest.

## 3. Results

A total of 2042 (97%) products representing seven food categories were included in the analysis (Figure 2). The food categories with more products were non-alcoholic beverages with added sugar (27%) and bakery products (cakes, sweet biscuits, and pastries) (26%).

### 3.1. Nutritional Content 

Table 1 shows the nutritional content of food and beverage products. The categories with the highest energy density (≥400 kcal/100 g) were savory snacks (500 kcal/100 g), cakes (440 kcal/100 g), sweet biscuits, and pastries, and confectionary products (400 kcal/100 g). Total fat was higher in savory snacks (26.0 g/100 g) than in cakes, sweet biscuits, and pastries (16.7 g/100 g). These two groups and confectionary products had the highest content of saturated fat (>6.5 g/100 g). Total added sugars content (>25 g/100 g) was highest in confectionery products (53.2 g/100 g), breakfast cereals (28.8 g/100 g), cakes, sweet biscuits and pastries (25.9 g/100 g). Finally, sodium (>300 mg/100 g) was the highest in savory snacks (571 mg/100 g), breakfast cereals (367 g/100 g), and cakes, sweet biscuits and pastries (333 mg/100 g). Since trans fats content was 0 grams in products, trans fats were excluded from all the statistical analysis. 

### 3.2. Declaration of Critical Nutrients

Table 1 shows the proportion of food products that declare critical nutrients in their NFP. Total fat (78%), saturated fat (74%), and sodium (77%) were less-commonly reported on packages of cakes, sweet biscuits, and pastries than on those of other food categories. Total added sugars were the least often reported on packages of savory snacks (67%). The least reported critical nutrient in all food categories were total added sugars (79%) and saturated fat (83%). Energy (90%) and total fat (90%) were the nutrients most reported in all food groups, and sugar (79%) was the least reported critical nutrient.

### 3.3. Claims

Table 2 shows claims included on FoP. Across all food categories, nutrition claims were more prevalent than health claims (43% of products with at least one nutrition claim versus 9% with at least one health claim). Only 7% of sweetened milk beverages and drinks without added sugar and 15% of breakfast cereals did not carry any nutrition claim on the FoP. Regarding health claims, sweetened milk beverages (53%) and breakfast cereals (19%) were the categories with most products with at least one health claim on the FoP. Products with the lowest number of nutrition and health claims were cakes, sweet biscuits, and pastries (13% and, 3% respectively) and confectionary products (24% and 2% respectively). 

Overall, nutrient content claims referring to vitamin, minerals, and sugar were most prevalent (44%) (Figure 3). Moreover, 62% of sweetened milk beverages and 33% of breakfast cereals displayed more than four nutrition or health claims in their FoP. General health claims were the most common type of health claim on the FoP (73%). Claims that referred to a “healthy product” that use words like “goodness”, “nutritious” or “super” to describe the product, and claims that referred to low glycemic index or energy density were the most commonly used general health claims (50% and 15%, respectively.

Table 3 shows a comparison between energy content and critical nutrients between products with and without at least one claim in the FoP. Energy and critical nutrients were significantly higher in products that do not include any nutrition or health claim.

### 3.4. Promotional Marketing Strategies

The presence of promotional characters on the FoP was highest in breakfast cereals (45%) and savory snacks (43%) and lowest in non-alcoholic drinks without sugar and with added sugars (3 and 11% respectively). The presence of premium offers on FoP was highest in breakfast cereals (15%), drinks without added sugar did not include premium offers in their FoP (Table 2).

Promotional characters commonly displayed on these products were cartoons and company-owned characters (74%), promotional characters for kids (13%). Premium offers commonly used were collectible gifts (34%) and contest (32%) (for example, keep two packages and change for one product or active codes).

Table 3 shows that the content of critical nutrients was significantly higher in products that displayed at least one promotional character on their FoP, compared to products that did not. Sodium (*p* = 0.00) and added sugar content (*p* = 0.01) were significantly higher in products that included at least one premium offer, compared to products that did not include premium offers in the FoP. No significant differences were found in the rest of the nutrients.

## 4. Discussion

This study is the first in Costa Rica to determine differences in critical nutrient and energy content of snacking products according to the presence or absence of specific claims and MS on the FoP labels of ultra-processed snack food products from Costa Rica.

Our main finding is that the nutritional profile of products that included promotional characters and premium offers in their FoP were less healthy than those products that did not include them. Similar results were found in other studies in which they concluded that products that try to engage children with their packaging design are less nutritious than foods that do not. Meanwhile, product packages that suggest nutritional benefits with their claims have more nutritious content [25,39,40], and nutrient profiles were healthier in products that displayed nutrition and health claims in their FoP. These results are similar to a Canadian study, in which they found that nutrient profile was poorer in those foods and beverages that did not include nutrition claims [41]. 

In our study, marketing strategies and claims were displayed on all ultra-processed snack food products. Our findings are consistent with recent studies performed in New Zealand, Ireland, United Kingdom, Brazil, showing widespread use of claims in food products targeted to children. Findings suggest that a lack of regulations could be the reason for this situation [42,43,44,45]. 

A lack of critical nutrient declaration on the most purchased products is a matter a concern, as an excess of critical nutrient consumption including sodium, fat, energy and added sugars are strongly associated with the onset of obesity and non-communicable diseases such as hypertension, diabetes and cardiovascular diseases [46,47,48,49,50]. 

In 2011, Blanco-Metzler analyzed the display of energy, total fat, saturated fat, and sodium on prepackaged food marketed in Costa Rica. From 2011 to 2015 the percentage of critical nutrients reported in the NFP has increased by 25% according to our study and the study mentioned before [51]. 

The proportion of sweetened milk beverages, breakfast cereals, drinks with and without added sugars with nutrition claims were higher than for the other food categories. Additionally, breakfast cereals and sweetened milk beverages were the categories with the highest number of claims (nutrition and health claims) on the FoP label. Nutrient content claims were the most popular type of claim use in food products. Findings from a recent study in Canada show that nutrient content claim was the most common claim used [41]. Additionally, sweetened milk beverages was the category with the highest number of health claims. Similar results were observed in previous studies in Costa Rica [21,51]. Consistent with other studies, products with at least one nutrition or health claim had a lower content of critical nutrients compared to those without such claims [43].

Breakfast cereals and savory snacks were the food categories with the highest proportion of promotional strategies on FoP labeling. These data are relevant because according to our nutrient profiles analyses, breakfast cereals is the food category with a higher content of added sugars and sodium. These results are consistent with a New Zealand study that analyzed the nutritional composition of some food categories and found that 58% of breakfast cereals for children were classified as less healthy [52]. Similarly, the savory snacks category is one of the categories with a higher content of saturated fat and sodium, implying that products with higher use of MS are less healthy. Additionally, it is well known that these strategies directly affect the selection of products in children [23,24,25,26,27]. Experimental studies showed that the presence of promotional characters could increase children’s appetite, preference for, choice of, and intake of less healthy foods [24]. Regarding parents’ influence, an experimental study found that parents believe the product was healthier based on the presence of health claims [18].

In Costa Rica, the food industry follows the Central American Technical Regulation of Nutrition Labeling, which is based on the CODEX Alimentarius voluntary guidelines [31]. However, this regulation has some weaknesses as reporting of critical nutrients such as added sugars, sodium, or fat is not mandatory, except when a health or nutrition claim is displayed [30]. In addition, there is no regulation about the veracity of claims included on food product packages; therefore, many health claims are confusing and can motivate consumers to buy some products [50]. 

This evidence suggests the importance of including a national regulation. In Latin America, some countries have already introduced regulations for unhealthy food marketing targeted to children. The Law of Nutritional Composition of Food and Advertising from Chile could be a model for Costa Rica. This regulation classifies food products as high or low content of calories, saturated fat, added sugars, and sodium content, and restricts all kinds of promotional strategies of food products with a high content of those critical nutrients. Another advantage of the law is that all food products exceeding these limits have to include a black and white nutritional warning system in the FoP [28,53]. Similar examples are being followed by Uruguay and Peru [54,55].

## 5. Conclusions

This study has some limitations. Firstly, the information about food products came from only one supermarket; nonetheless, it is one of the most popular and largest supermarket chains in the country. Another limitation is that there was not a nutrient profile, with established limits for the food categories selected, in order to classify them as healthy and unhealthy. The nutrient profile from PAHO and WHO classified all products as unhealthy, so we were unable to make comparisons between healthier and unhealthier products within the same categories. Lastly, some photographs of food packages were incomplete and therefore, the package could not be analyzed thoroughly. 

Our study provides evidence of the abundant use of marketing strategies in ultra-processed food products commonly targeted to children and adolescents in Costa Rica. Additionally, our results highlight the use of promotional characters and premium offers as MS in the FoP of snacking products, particularly in foods and beverages with poorer nutritional profiles. These results can be used for taking action against obesity in the country. Further, there is an urgent need for health authorities to develop regulations that restrict or ban the use of these strategies in unhealthy snacks and beverages products. These regulations have to be useful for verifying the accuracy and veracity of health and nutrition claims displayed on labels. Finally, the inclusion of an evidence-based FoP labelling system, easy to understand for the population, should be included in legislation to generate healthier food environments and contribute to the prevention of childhood and adolescent obesity in Costa Rica. 

## Figures and Tables

**Figure 1 nutrients-11-02738-f001:**
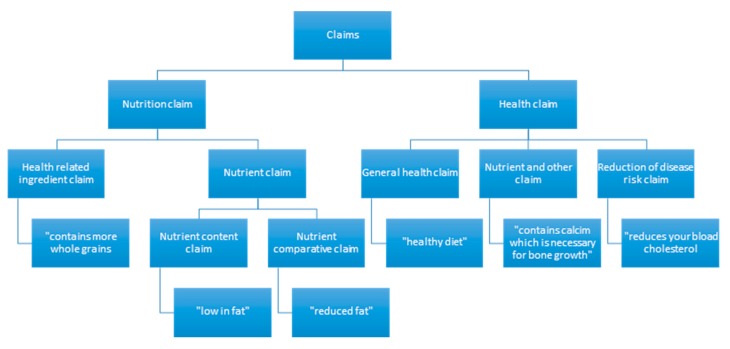
INFORMAS taxonomy for the classification of nutrition and health claims [38].

**Figure 2 nutrients-11-02738-f002:**
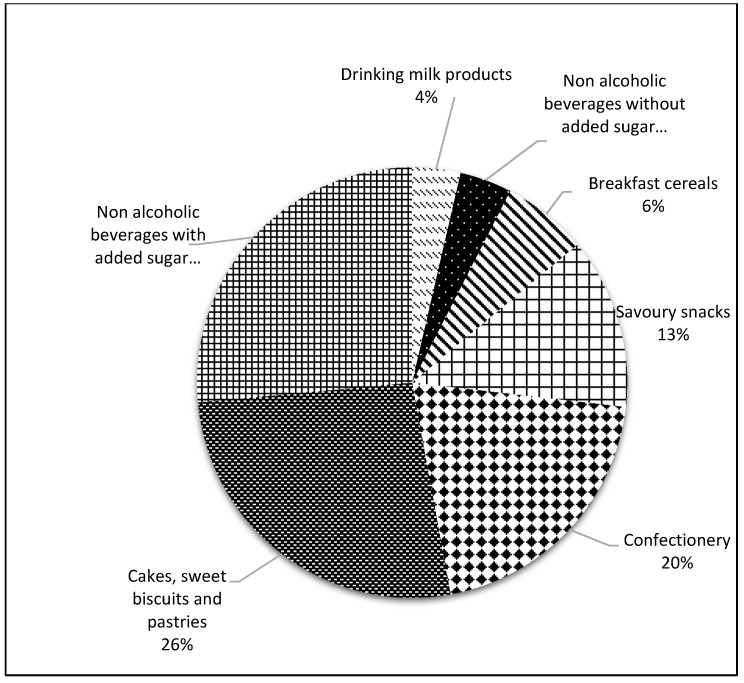
Overall proportion of ultra-processed food products commonly consumed as a snack by children and adolescents, according to the INFORMAS food categories classification (34), Costa Rica 2015 (*n* = 2402).

**Figure 3 nutrients-11-02738-f003:**
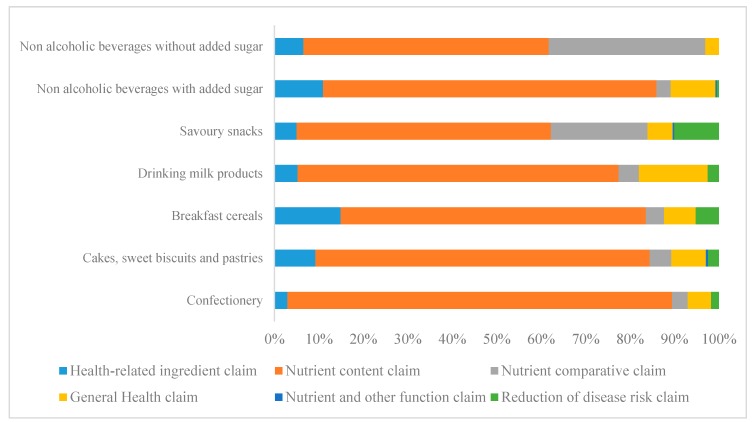
Percentage of different types of nutrition and health claims by food category included in FoP of ultra-processed food products commonly consumed as a snack by children and adolescents, Costa Rica 2015.

**Table 1 nutrients-11-02738-t001:** Proportion of declared and content of critical nutrients ^a,b^ in ultra-processed food products commonly consumed as a snack by children and adolescents, Costa Rica 2015 (per 100 g/mL).

Food Categories ^c^	Energy		Total Fat		Saturated Fat		Trans Fat		Total Free Sugar		Sodium	
	*n* (%)	(kcal)	*n* (%)	(g)	*n* (%)	(g)	*n* (%)	(g)	*n* (%)	(g)	*n* (%)	(mg)
Confectionery	458 (94)	400 (337.0–506.9)	451 (92)	6.2 (0.0–27.8)	363 (74)	8.3 (0.0–16.3)	206 (42)	0 (0.0–0.0)	446 (91)	53.2 (39.0–61.2)	444 (91)	50 (0.0–100.0)
Cakes, sweet biscuits and pastries	493 (78)	440 (382.4–492.5)	493 (78)	16.7 (10.2–22.2)	475 (75)	6.5 (3.6–10.0)	474 (75)	0 (0.0–0.0)	458 (72)	25.9 (11.1–35.0)	486 (77)	333 (214.3–608.7)
Breakfast cereals	151 (98)	387 (366.7–400.0)	151 (99)	6.1 (3.3–9.1)	145 (94)	1.1 (0.0–3.0)	139 (90)	0 (0.0–0.0)	142 (92)	28.8 (16.7–34.3)	148 (96)	367 (126.8–486.8)
Drinking milk products	87 (100)	61 (52.9–66.7)	87 (100)	1.8 (1.3–2.4)	78 (90)	0.7 (0.2–1.0))	42 (48)	0 (0.0–0.0)	72 (83)	5.5 (4.0–8.3)	81 (93)	43 (35.4–66.7)
Savory Snacks	290 (94)	500 (461.0–535.7)	290 (94)	26.0 (20.0–32.1)	274 (89)	8.4 (4.0–12.0)	264 (85)	0 (0.0–0.0)	206 (67)	0.1 (0.0–3.6)	283 (92)	571 (321.4–815.8)
Beverages with sugar	593 (93)	39 (23.5–50.0)	589 (93)	0 (0.0–0.0)	573 (90)	0 (0.0–0.0)	569 (89)	0 (0.0–0.0)	512 (81)	7.8 (4.9–11.6)	548 (86)	8 (3.0–15.3)
Beverages without added sugar	94 (100)	2 (0.0–11.2)	94 (100)	0 (0.0–0.0)	88 (94)	0 (0.0–0.0)	87 (93)	0 (0.0–0.0)	72 (77)	0 (0.0–1.6)	85 (90)	10 (4.4–14.6)
Al food categories	2166 (90)	366.7 (46.0–468.7)	2155 (90)	3.7 (0.0–21.0)	1996 (83)	1.0 (0.0–8.3)	1781 (74)	0 (0.0–0.0)	1908 (79)	11.6 (3.6–37.2)	2075 (86)	70 (8.8–354.2)

^a^ IQR: interquartile range, ^b^ Data expressed in median and 25th–75th percentile. ^c^ INFORMAS food categories classification.

**Table 2 nutrients-11-02738-t002:** Proportion of products included claims and marketing strategies on the front of package of ultra-processed food products commonly consumed as a snack by children and adolescents, Costa Rica 2015.

Food Categoriesb ^a^	Front of Pack with at Least One					
	Nutrition Claim *n* (%)	Health Claim *n* (%)	Nutrition or Health Claim *n* (%)	Promotional Characters *n* (%)	Premium Offers *n* (%)	Promotional Characters or Premium Offers *n* (%)
Confectionery (*n* = 488)	119 (24)	12 (2) ^A^	119 (24)	100 (20) ^A^	12 (2) ^A^	107 (22) ^A^
Cakes, sweet biscuits and pastries (*n* = 634)	80 (13)	17 (3) ^A^	83 (13)	89 (14) ^A,B^	27 (4) ^A^	114 (18) ^A,B^
Breakfast cereals (*n* = 154)	131 (85) ^A^	30 (19) ^B^	131 (85) ^A^	73 (47) ^C^	25 (16)	85 (55) ^C^
Drinking milk products (*n* = 87)	81 (93) ^A^	46 (53)	85 (98) ^B^	22 (25) ^A,D,E^	3 (3) ^A^	23 (26) ^A,D^
Savoury snacks (*n* = 309)	142 (46)	42 (14) ^B^	148 (48)	134 (43) ^C,D^	8 (3) ^A^	136 (44) ^C,D^
Drinks with added sugar (*n* = 636)	402 (63)	74 (12) ^B^	427 (67)	67 (11) ^B,E,F^	11 (2) ^A^	74 (12) ^B^
Drinks without added sugar (*n* = 94)	87 (93) ^A^	6 (6) ^A,B^	87 (93) ^A,B^	3 (3) ^F^	0 ^B^	3 (3)
Total of products (2402)	1042 (43)	227 (9)	1080 (45)	488 (20)	86 (4)	542 (23)

^a^ INFORMAS food categories classification. ^A, B, C, D, E, F^ Percentages with the same superscript letters weren’t significantly different between the column percentages, based on Marascuillo procedure.

**Table 3 nutrients-11-02738-t003:** Energy and critical nutrient content ^a^ in 100 g of product with and without claims and marketing strategies on the front of the package of ultra-processed food products commonly consumed as a snack by children and adolescents, Costa Rica 2015.

Nutrients ^b^	Front of Pack with at Least One											
	Nutrition Claim		Health Claim		Nutrition or Health Claim		Promotional Characters		Premium Offers		Promotional Characters or Premium Offers	
	Yes	No	Yes	No	Yes	No	Yes	No	Yes	No	Yes	No
Energy	60.0 (24.4–396.3) *	427.0 (333.3–500.0)	65.2 (48.0–400.0) *	372.1 (45.8–475.0)	60.0 (24.4–394.5) *	431.9 (340.0–500.0)	380.0 (318.0–500.0) *	355.0 (40.0–466.7)	372.0 (347.8–406.2)	366.7 (45.1–470.0)	380.0 (326.7–480.2) *	352.9 (39.8–466.7)
Total fat	0.0 (0.0–8.7) *	14.5 (0.0–25.0)	1.8 (0.0–7.8) *	5.0 (0.0–21.6)	0.0 (0.0–8.0) *	15.0 (0.0–25.0	7.5 (0.0–25.0) *	2.6 (0.0–20.0)	4.0 (1.0–9.7)	3.8 (0.0–21.4)	7.0 (0.1–24.0) *	2.0 (0.0–20.1)
Satured fat	0.0 (0.0–2.0) *	5.4 (0.0–10.9)	0.3 (0.0–2.0) *	1.5 (0.0–8.9)	0.0 (0.0–2.0) *	5.7 (0.0–11.1)	3.3 (0.0–9.2) *	0.4 (0.0–7.9	1.2 (0.0–2.7)	1.0 (0.0–8.4)	2.3 (0.0–9.0) *	0.0 (0.3–8.0)
Free Sugars	6.5 (1.8–14.2) *	28.6 (7.7–50.0)	5.8 (2.4–13.3) *	12.5 (3.6–39.8)	6.6 (1.9–14.0) *	30.0 (7.9–50.0)	16.6 (3.9–40.6) *	10.7 (3.3–33.7)	30.0 (6.7–41.9) *	11.2 (3.5–35.0)	16.7 (4.2–41.9) *	10.4 (3.0–33.3)
Sodium	25 (5.2–271.4) *	138.9 (22.0–420.0)	45.8 (8.8–219.0) *	76.5 (8.8–375.0)	25.0 (5.5–260) *	150.0 (25.0–423.1)	222.5 (32.0–507.1) *	51.0 (7.1–308.4)	383.3 (120.0–666.7) *	66.7 (8.3–333.3)	265.5 (35.8–533.3) *	47.9 (6.7–284.3)

^a^ Data expressed in median and 25th–75th percentile. ^b^ A Man Whitney test (95%CI) was used to determine differences in nutritional content between products without claims and marketing strategies. * *p* < 0.05.

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
