# Peer review of "Nutritional Content According to the Presence of Front of Package Marketing Strategies: The Case of Ultra-Processed Snack Food Products Purchased in Costa Rica"

_nutrients, 2019, doi:10.3390/nu11112738_

Round 1
Reviewer 1 Report
This paper deals a very important topic on the field of public health policies, especially those instruments which fight against obesity such as nutritional and health claims. Also shows an interesting insight of how other regions with high prevalence of obesity deals with this pandemic. However, the paper needs improvements on it structure and an extensive editing of the English language and style before published it. Moreover, although the aim of the paper is analyze the nutritional quality of ultra-processed food product, the methodology used it in this study do not allows assess this issue.
Following I describe the improvements required.
Abstract
The abstract must follows the guidelines for authors of the journal. Abstract must describe in few lines the background of the topic addressing on the paper and the purpose of the study. Authors must avoid write the abstract with headings.
Introduction
The introduction is not clear and need be reorganized.
I suggest to the authors include more data about the prevalence of obesity on America and especially Costa Rica.
Topics mentions on lines 42 to 54 are not relevant and authors only briefly mention the impact of marketing strategies on child preference, which is more relevant to the purpose of the paper.
Follow to these paragraphs, the author can describe the public health context and the regulatory background in Costa Rica. In this paragraph, authors should describe all type of claims. This is important, because the readers of other regions can compare with their own regulatory background.
Finally, authors can present the main purpose of the study.
Authors must cite properly, in lines 60, 67, 77 cites were omitted.
Materials and Methods
Although the authors mentions that the INFORMAS methodology was follow in this study, the methodology section cannot be reproduce by other researchers. Methodology section does not describe in detail the procedure using to obtain the data, to treat the data and how the data base was constructed. Authors do not mention the exclusion and inclusion criterion of food category.
Authors must be explain why in methodology section the sample is 7953 food products but then the sample was reduced to 2457.
In nutritional content, authors must explain the criterion using to calculate the proportions and describe the measure to compare with the sample. Also, the authors should mention if the nutritional content was check in laboratory or other method.
In line 134, authors indicate nutritional content but do not describe the procedure follows to determine it.
Result
Authors said they exclude the trans-fat of the statistical analysis; hence this element should be removed from the tables.
In general, authors must be expanding the explaining of the tables. The description of the result is very bare.
I suggest that the table presents only proportions; the n of each food category can be presented on the row title.
The subheading 3.2 should be present before table 1.
On subheading “3.3. Claims”, the percentage do not correspond with those on table 2.
Please, check the sentence in line 194 because this is redundant.
The category mention on figure 2 was not described on methodology section.
Please check line 222.
Respect to the nutritional content analysis, this measure wasn’t defining it before in methodology section. So, I cannot assess if this analysis was well conducted. Moreover, I do not find any sense compare nutritionals profiles between food products with and without nutritional and health claims. Considering that the authors mention that in Central American regulation of the nutritional and health claims is lax, I found more interesting either compare between laboratory data (obtaining of the sample) and theoretical data (nutritional table of the sample) or identify if the sample use the nutritional/health claim according to the normative (RTCA 67.01.60:10 Etiquetado Nutricional de Productos Alimenticios Preenvasados para Consumo Humano para la Población a partir de 3 años de Edad).
Authors did not include the regulation “RTCA 67.01.60:10 Etiquetado Nutricional de Productos Alimenticios Preenvasados para Consumo Humano para la Población a partir de 3 años de Edad”. This regulation is important because it defines the authorized nutritional and health claim in Central America. I suggest include the regulation RTCA 67.01.60:10 in the introduction and methodology section.
Discussion
Discussion must be summarizing in few lines the main aim of the study. I suggest include a short paragraph (2-3 lines) that remember to the readers the purpose of the study and then comment the results.
In light of present the social, public health policies and marketing implications of this study. It is relevant to compare the present study with similar studies in America or Europe.
Reviewer 2 Report
Thank You very much for the possibility to become familiar with an interesting article. It is well-written and has a research character. However, because the best is the enemy of the good, I suggest some improvements in that text.
First of all, in the introduction, the importance of the problem should be emphasised, and the aim of the research should be clearly stated. The introduction should also inform about the sources of collected information and briefly present the methodology of the study. The article should also contain hypotheses or at least research questions. After the introduction, in the "Literature review" section, there should be a reference to the current state of knowledge on the topic taken. I suggest a wider description of the way of selection of the examined sample. In my opinion, it should be emphasised if the selection was intentional or random. Even with the most accurate selection of the data analysis methods, the results obtained from non-random samples should be treated with caution. It should be also explained what kind of research method and technique was used. Why did the Authors consider it to be suitable? The size of the measurement error should be added. The explanation of the size of the sample should be expanded. Why does the sample have this particular size? The "Conclusions" section should be expanded. It should be indicated what the research brings to science and what are its application values. The Limitations of research should be placed in the Conclusions section rather than in the Discussions section.Author Response
Please see the attachment.

Round 2
Reviewer 1 Report
When I suggest review the English language and style I refer to improve the style, particularly. In the former version it was very hard follows and understands the principal point of the study. However, in the present version it was easier to read and understand the paper.
In general, the present paper’s version answers to all my questions and suggestions. But, some few typos must be fixed:
Please check the line 90 and 100
Please check the line 456, and the title of figure 1, the number sample is wrong. There must be 2402 not 2042.
Please check the reference section. Some references cited in text do not correspond with number in reference section.
Finally, in your documents ‘responses from authors’, tables’ row 21. I think there is some confusion with my suggestion. The normative ‘RCTA 67.01.07:10 Etiquetado General de los alimentos previamente envasados’ refers to general guidelines but the ‘RCTA 67.01:60:10 Etiquetado nutricional de productos alimenticios preenvasados para consume humano para la población a partir de 3 años de edad’ is more specific and stablish some requirements to include nutritional and health claims as labelling. I wonder if in only the former regulation is in force in Costa Rica.
